# From musk to body odor: Decoding olfaction through genetic variation

Bingjie Li[1,2☯], Marissa L. Kamarck[3,4☯], Qianqian Peng[1☯], Fei-Ling Lim[5], Andreas Keller[6], Monique A. M. Smeets[7], Joel D. Mainland[3,4]*, Sijia Wang[1,8]*

1 CAS Key Laboratory of Computational Biology, Shanghai Institute of Nutrition and Health, University of Chinese Academy of Sciences, Chinese Academy of Sciences, Shanghai, China, 2 Department of Skin and Cosmetics Research, Shanghai Skin Disease Hospital, School of Medicine, Tongji University, Shanghai, China, 3 Monell Chemical Senses Center, Philadelphia, Pennsylvania, United States of America, 4 Department of Neuroscience, University of Pennsylvania, Philadelphia, Pennsylvania, United States of America, 5 Unilever Research & Development, Colworth, United Kingdom, 6 Laboratory of Neurogenetics and Behavior, The Rockefeller University, New York, New York State, United States of America, 7 Unilever Research & Development, Rotterdam, The Netherlands, 8 Center for Excellence in Animal Evolution and Genetics, Chinese Academy of Sciences, Kunming, China

☯ These authors contributed equally to this work.
* wangsijia@picb.ac.cn (SW); jmainland@monell.org (JDM)

**Data Availability Statement:** Individual-level genotype and phenotype data are not publicly available owing to them containing information that could compromise research participant privacy or informed consent. Interested researchers would be

## Abstract

The olfactory system combines input from multiple receptor types to represent odor information, but there are few explicit examples relating olfactory receptor (OR) activity patterns to odor perception. To uncover these relationships, we performed genome-wide scans on odor-perception phenotypes for ten odors in 1000 Han Chinese and validated results for six of these odors in an ethnically diverse population (n = 364). In both populations, consistent with previous studies, we replicated three previously reported associations (β-ionone/OR5A1, androstenone/OR7D4, cis-3-hexen-1-ol/OR2J3 LD-band), but not for odors containing aldehydes, suggesting that olfactory phenotype/genotype studies are robust across populations. Two novel associations between an OR and odor perception contribute to our understanding of olfactory coding. First, we found a SNP in OR51B2 that associated with trans-3-methyl-2-hexenoic acid, a key component of human underarm odor. Second, we found two linked SNPs associated with the musk Galaxolide in a novel musk receptor, OR4D6, which is also the first human OR shown to drive specific anosmia to a musk compound. We noticed that SNPs detected for odor intensity were enriched with amino acid substitutions, implying functional changes of odor receptors. Furthermore, we also found that the derived alleles of the SNPs tend to be associated with reduced odor intensity, supporting the hypothesis that the primate olfactory gene repertoire has degenerated over time. This study provides information about coding for human body odor, and gives us insight into broader mechanisms of olfactory coding, such as how differential OR activation can converge on a similar percept.

able to obtain the GWAS summary statistics derived by following the detailed methods described in the manuscript. The dataset of GWAS summary statistics were deposited in the National Omics Data Encyclopedia (http://www.biosino.org/node/, Project ID: OEP001806). Data usage shall be in full compliance with the Regulations on Management of Human Genetic Resources in China. All other data are contained in the article file and its supplementary information.

**Funding:** This work was supported by the Strategic Priority Research Program (Grant No. XDB38020400), the National Key Research and Development Project (Grant No. 2018YFC0910403), the National Natural Science Foundation of China (Grant No. 91631307), Shanghai Municipal Science and Technology Major Project (Grant No.2017SHZDZX01) to SW, CAS Youth Innovation Promotion Association (Grant No. 2020276) to QP, the National Institutes of Health (Grant R01 DC013339) to JDM, and in part by the National Center for Advancing Translational Sciences Clinical and Translational Science Award program (grant UL1 TR000043) to AK. A portion of the work (validation study) was performed at the Monell Chemosensory Receptor Signaling Core, which was supported in part by the National Institute on Deafness and Other Communication Disorders (Core Grant P30 DC011735) to JDM. The discovery study was funded in part by Unilever R&D (the Netherlands) to SW. Unilever employees M.A.M.S. and F.-L.L. are co-authors who contributed in study conceptualization and writing (review & editing). M.A.M.S. also provided resources (odor sticks for the discovery study). Other funders had no role in study design, data collection and analysis, decision to publish, or preparation of the manuscript.

**Competing interests:** I have read the journal's policy and the authors of this manuscript have the following competing interests: The study was funded in part by Unilever R&D (the Netherlands). M.A.M.S. and F.-L.L. are employees of Unilever. The other authors declare that they have no competing interests.

## Author summary

Although genetic diversity in the olfactory receptor repertoire contributes to variation in odor perception, we have few explicit predictions relating variation in a specific OR to perception. Here, we performed genome-wide scans on odor-perception phenotypes for ten odors in 1000 Han Chinese and validated results for six of these odors in an ethnically diverse population (n = 364). We identified novel receptors for musk and human body odor that have implications for how structurally different molecules can have similar odors. Summarizing all the published genetic variation that associates with odor perception, we found that individuals with ancestral versions of the receptors tend to rate the corresponding odor as more intense, supporting the hypothesis that the primate olfactory gene repertoire has degenerated over time. This study of olfactory genetic and perceptual variation will improve our understanding of how the olfactory system encodes odor properties.

## Introduction

Every individual experiences smell in their own unique way–variation in odor perception can range from specific anosmias, where an individual completely lacks the ability to perceive a particular odorous compound, to differences in individual experience of quality, pleasantness, and/ or intensity of odors [1]. Comparing this perceptual variability with genetic variability allows us to identify the role of single odorant receptors in the perceptual code. Progress in sequencing technology and increased access to previously genotyped cohorts has enhanced our ability to uncover the genetic components underlying differences in odor perception.

Olfactory receptors (ORs), the family of proteins responsible for detection of odor compounds, have a high level of genetic variation relative to other proteins [2–4]. Of the 800 olfactory receptor genes, only about 400 are intact, and, on average, approximately 30% of OR alleles will differ functionally between two people [5]. Even within the set of intact genes, a genetic variant can alter function of a single OR and thereby alter perception of an odor. To date, there are 15 cases where perceptual variability of an odor correlated with a genetic variant in a receptor that responds to the odor in a cell-based assay [5–12], and 13 further cases with strong genetic evidence, but no supporting evidence from cell-based assays [10,11,13].

Here, we utilize the same strategy of correlating perceptual and genetic variation, but with three improvements: 1. Using a larger population to increase power, 2. Conducting genetic analysis in an understudied population (Han Chinese), as opposed to previous studies that have been largely conducted in Western (majority Caucasian) populations, and 3. Validating the results using an independent population and different methodology, demonstrating the robustness of the finding.

In this study, we tested a Han Chinese population (n = 1000) alongside a smaller validation cohort (n = 364) of a Western population, using odors that have unexplained variability in perception–Galaxolide, trans-3-methyl-2-hexenoic acid (3M2H), and aldehydes–as well as a set of odors with previously described associations between perceptual variability and genetic variants.

### Galaxolide: A musk compound

The olfactory literature contains a number of examples of compounds with very different structures but similar odors [14]. The perceptual category of musks is perhaps the most

striking example. Compounds in five different musk structural classes–macrocyclic, polycyclic, nitro, steroid-type, and straight-chain (alicyclic)–all have a similar perceptual quality described as sweet, warm, and powdery [15]. The simplest explanation is that all musk structures activate one receptor or one common subset of receptors that in turn encodes the perceptual "musk" quality; however, evidence suggests coding of this percept may be more complex. Individuals can have specific anosmias to one or some, but not all musks [16,17], suggesting that there is not a single common coding mechanism.

In this study, we examined Galaxolide, a musk compound with a characterized specific anosmia [16,17]. Galaxolide does not activate OR5AN1, which was shown to be critical for the perception of other musk compounds in mice [18]. The structural variance and common percept amongst musk compounds allows us to examine different coding mechanisms that are central to our understanding of how receptor activation relates to odor perception.

### 3M2H: A body odor contributor

All mammals use chemosensation as a means of intra-species communication, but the mechanism of chemosensory communication amongst humans is largely unknown. The growing evidence for chemical communication between humans suggests that body odor is of particular importance, as it may be processed differently in the brain than other odors [19] and may influence various social behaviors including kinship recognition, mate selection [20], and fear priming [21]. Although 3M2H is only one of ~120 compounds [22] that comprise body odor, it is an "impact odor", meaning that it carries the characteristic scent of body odor [23]. Furthermore, almost 25% of the population has a specific anosmia to 3M2H [23–26], but this anosmia has not been connected to any olfactory receptor. Identifying receptors responsible for perception of 3M2H and body odor may have implications for social communication, malodor prevention, and receptor coding mechanisms for conspecific odors.

### Replicating odor associations

Previous publications have implicated OR genetic variation in perception of specific odors. To examine if these associations are robust and consistent across populations, we measured responses to β-ionone [9], androstenone [6,10], cis-3-hexen-1-ol [8–10,27], and caproic acid [10].

### Testing aldehydes in different populations

Aldehydes have been shown to vary perceptually across demographic groups such that self-reported Asian populations rate aldehydes as more intense than Caucasian populations [28], but no specific genetic variants or receptors have been implicated. To assess the genetic underpinnings of aldehyde preferences in the Han Chinese population, we tested two monomolecular aldehydes: decyl aldehyde [28] and galbanum oxathiane, alongside two fragrance mixtures used in home care products: MixA, which has high levels of aldehydes and is relatively unpopular in Asia, and MixB which has low levels of aldehydes and is popular in Asia.

### Results

To discover genetic variants related to differences in odor perception, we examined how genetic variation correlated with olfactory phenotypes in two cohorts. The discovery cohort consisted of 1000 (369 male) Han Chinese participants, and the validation cohort consisted of 357 (161 male) participants collected in New York City. We conducted a principal component analysis (PCA) and genetic distance analysis of all identified genetic variants to confirm

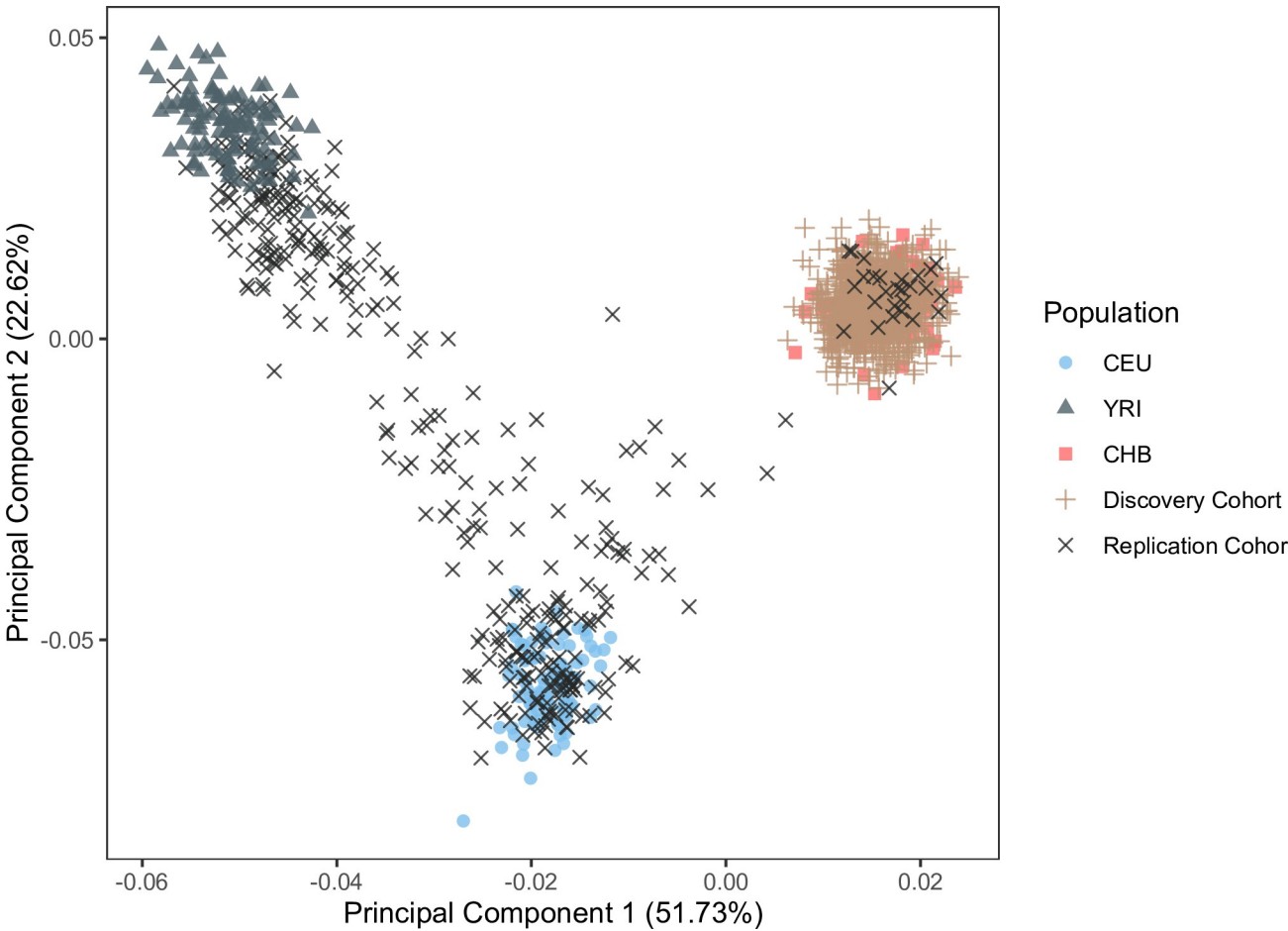

**Fig 1. Population structure analysis reveals relative homogeneity of the discovery population (tan plus sign) compared to the validation population (black x) (p<2.2x10$^{-16}$).** Shown are the first two principal components calculated from all variants genotyped in both the discovery and validation cohorts. Representative populations from the 1000 Genomes Project: Han Chinese in Beijing (CHB, n = 97; red square), Utah residents with Northern and Western European ancestry from the CEPH collection (CEU, n = 86; blue circle), and Yoruba in Ibadan, Nigeria (YRI, n = 88; black triangle) are plotted for context. The discovery population overlapped with the CHB population (mean distance to CHB = 0.001, CEU = 0.07, YRI = 0.07).

relative homogeneity/heterogeneity of our discovery and replication populations, respectively. The first two principal components explained 52% and 23% of the genetic variance. We confirmed that the discovery population overlapped with the Han Chinese population (CHB) from the 1000 Genomes Project [29] (mean distance to CHB = 0.001, CEU = 0.07, YRI = 0.07), whereas the validation study population was distributed between different super-populations (mean distance to CHB = 0.05, CEU = 0.05, YRI = 0.05). The mean distance between any two participants within a study cohort is smaller in the discovery population (mean distance = 0.007) than in the validation population (mean distance = 0.047), confirming that the discovery population is more homogeneous than the validation population (p<2.2x10$^{-16}$; Fig 1).

Participants from both cohorts rated intensity and pleasantness of all odors on a 100-point scale. The discovery cohort had 10 odors presented at a single concentration. Most participants performed each olfactory rating task once, but for each odor a set of 100 participants rated the odor twice throughout the session (test-retest r = 0.75). The validation cohort tested 6 of the 10 odors in the discovery cohort, some of which were presented at two concentrations (high/ low). Each participant rated all odors twice throughout the session (r = 0.69) (S1 Fig).

In both cohorts, we normalized participant ratings by ranking across odors by intensity and pleasantness. For the discovery cohort, we performed genome wide association analysis for 20 olfactory phenotypes (S1 Data). We identified novel genetic variants (Fig 2) associated with the intensity rankings of Galaxolide and trans-3-methyl-2-hexenoic acid (3M2H) that explain 13.27% and 4.12% of the phenotype variance, respectively. We can compare this to the maximum expected values provided by heritability analysis, which estimates 33% (Galaxolide) and 25% (3M2H) of phenotypic variance is attributable to genetic variation (S1 Table). In addition, we replicated published associations for β-ionone, androstenone, and cis-3-hexen-1-ol (Fig 2). The validation cohort replicated novel associations identified in the discovery study

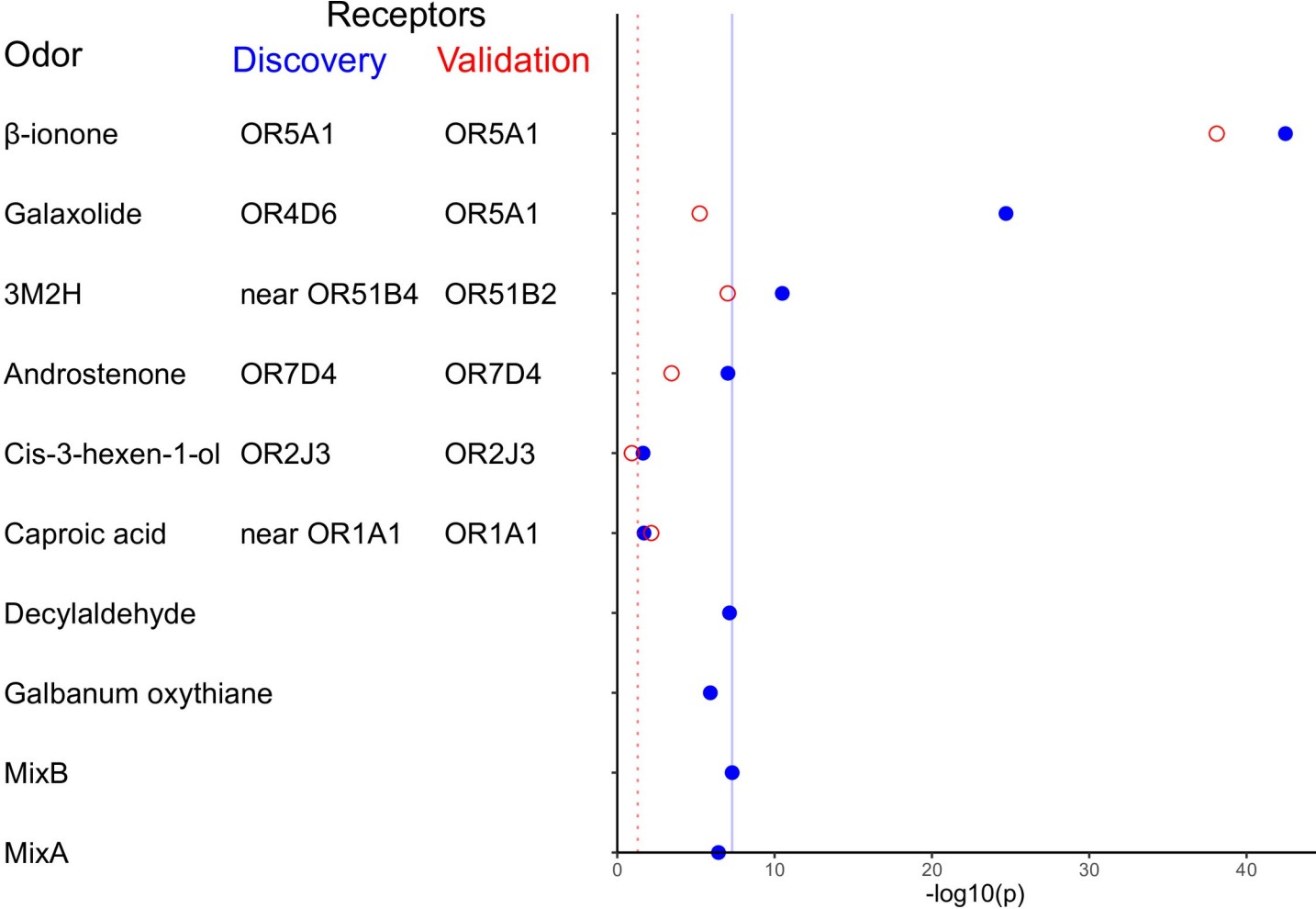

**Fig 2. Top associations between genetic variation and odor intensity perception.** Each row represents the top SNPs associated with the odor intensity phenotype in the discovery cohort (blue filled circles) and the replication cohort (red open circles). Listed next to each odor is the nearest gene to the top SNP for each cohort. There were two novel associations that reached genome-wide significance in the discovery cohort (p < 2.5x10⁻⁹, solid blue line): Galaxolide/rs1453541(M263T) and rs1453542 (S151T) (p<2.4x10⁻²⁵, p<3.0x10⁻²⁵) and 3M2H/rs3898917 (p<1.9x10⁻¹¹). The discovery study replicated the associations from the literature (p<0.05, dotted red line) for β-ionone/rs6591536 (D183N) (p<5.5x10⁻⁴²), androstenone/rs61729907 (R88W) and rs5020278 (T133M) (in both cases, p<9.3x10⁻⁸), and cis-3-hexen-1-ol/rs28757581 (T113A) (p<0.02). The discovery cohort was unable to impute the region around OR1A1 for replication of the caproic acid association. Other than β-ionone, no replication odors had associations that reached genome-wide significance. For this set of replication odors (β-ionone, androstenone, cis-3-hexen-1-ol, and caproic acid), the association shown is the top association from the LD-band surrounding the previously identified SNP. In the validation cohort (open red circles), we tested associations for the significant SNPs and surrounding LD-bands (±200kb) from the discovery study and previous literature. There were four significant associations in the validation study (p<0.05; red dotted line): β-ionone/rs6591536 (D183N) (p < 7.8x10⁻³⁹), Galaxolide/rs591536 and rs7941591 (p<5.8x10⁻⁶), 3M2H/rs10837814 (L143F) (p<9.6x10⁻⁸), and androstenone/rs61732668(P79L) (p<3.5x10⁻⁴). The association for caproic acid/rs17762735 (p<6.9x10⁻³) is significant, but in the opposite direction predicted by the previous study.

(Galaxolide, 3M2H) as well as published associations (β-ionone, androstenone, cis-3-hexen-1-ol). In both cohorts, all the genetic variants that were significantly associated with pleasantness were also significantly associated with intensity perception (S1, S2, and S3 Data).

## OR4D6 variant alleles M263T and S151T are associated with a decrease in Galaxolide intensity

In the discovery study, Galaxolide intensity perception was associated with an OR locus in chromosome band 11q12.1 (Fig 3A and 3B). The two peak variants in open reading frames were both missense single nucleotide polymorphisms (SNPs) in OR4D6 (Fig 3C): M263T (rs1453541, p<2.42x10$^{-25}$) and S151T (rs1453542 p<3.03x10$^{-25}$). The validation study

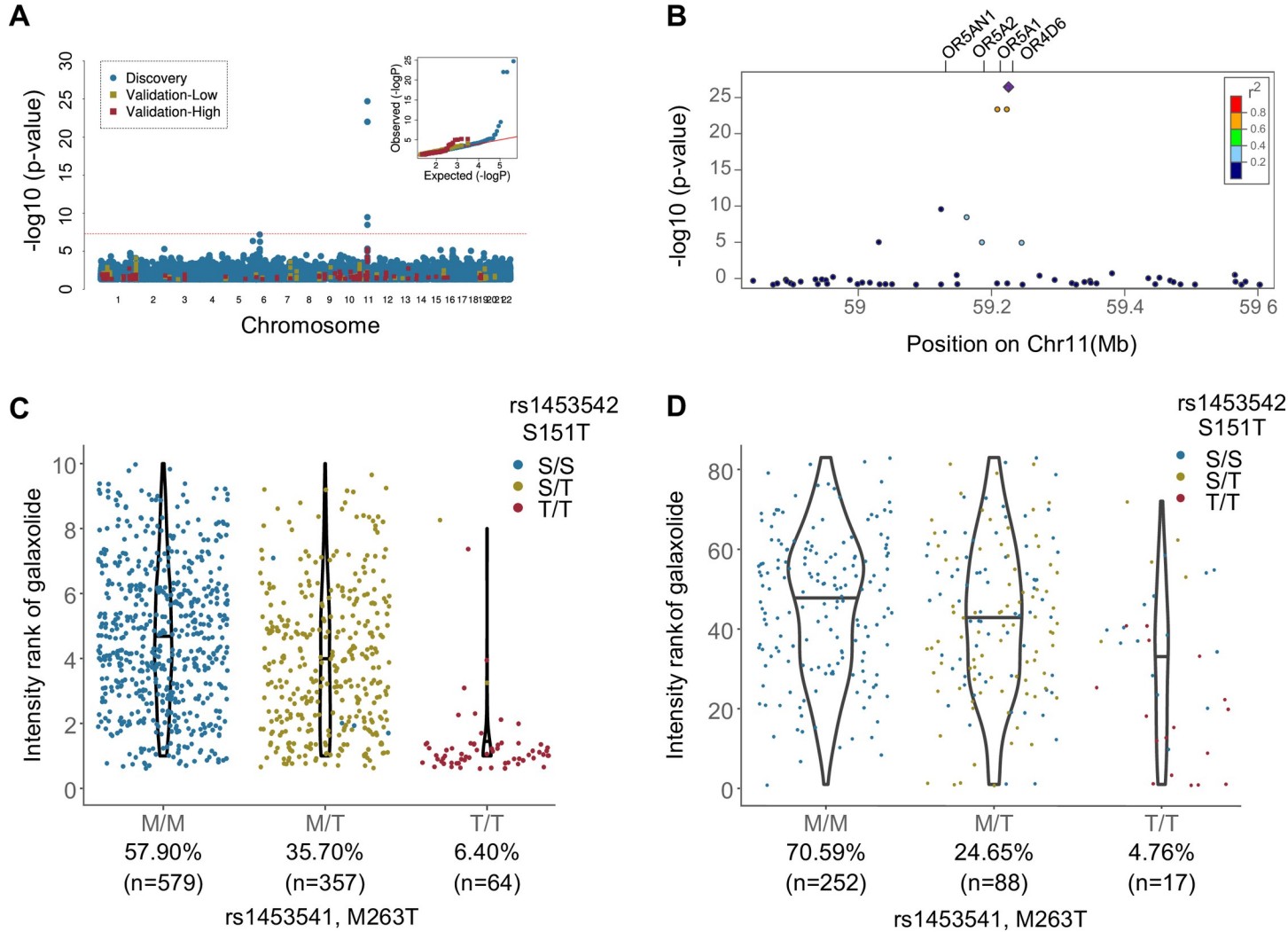

**Fig 3. Galaxolide perception is associated with variation in chromosome band 11q12.1 in both cohorts**, as shown by **A)** a Manhattan Plot of associations with the discovery study in blue and validation study in red (high concentration) and yellow (low concentration). The red line indicates the threshold for genome-wide significance (p<5x10$^{-8}$). Inset: QQ plots from the discovery (blue) and validation ([high] = red, [low] = yellow) cohorts (Genomic Lambda: discovery = 1.02; validation = 0.90) show appropriate control for inflation due to population structure. **B)** The regional plot of discovery study associations indicates both the significance level and the recombination rate at the OR4D6 LD-band. Genetic variation in OR4D6 affects the perceived intensity of Galaxolide in **C)** the discovery cohort and **D)** the validation cohort (high concentration). The x-axis is ordered left-to-right with increasing number of variant alleles for the M263T variant, for population frequency of M263T indicated below the genotype. The points colored by the S151T genotype suggest that in the validation cohort, S151T is driving the Galaxolide anosmia phenotype exhibited by those homozygous for the variant (T/T).

confirmed that both OR4D6 SNPs correlated with the intensity of the higher of the two tested concentrations of Galaxolide (M263T $p<9.08 \times 10^{-6}$, S151T $p<1.02 \times 10^{-5}$, Fig 3D). The two SNPs are in high linkage disequilibrium (LD): the variant allele of S151T is always co-inherited with the variant allele of M263T (S2 Table).

We examined the associations between Galaxolide and SNPs in other reported musk-activated ORs, including OR5AN1 (activated by muscone), OR5A2 (activated by all musk compound families), and OR1A1 (activated by nitro musks) [18,30,31]. Of these ORs, only SNPs in OR5AN1 and OR5A2 were significantly associated with Galaxolide ($p = 3.25 \times 10^{-8}$ and $p = 4.70 \times 10^{-17}$, respectively; S3 Table). Since both of these SNPs are in strong LD with the novel signal discovered in OR4D6, we performed further analysis controlling for the top associated SNP in OR4D6, which did not reveal any signal reaching genome-wide significance. Additionally, only OR4D6 is in the credible set of the fine mapping analysis (S4 Table).

The leading role of OR4D6 is also supported by the meta-analysis of both cohorts where the two OR4D6 SNPs were the top two associations with Galaxolide intensity (M263T $p<3.81 \times 10^{-29}$, S151T $p<5.42 \times 10^{-29}$; S2 Data). The meta-analysis revealed no significant associations with Galaxolide pleasantness. Based on the evidence from the above analyses we examined the effect size of the two SNPs in OR4D6 on Galaxolide intensity ratings. The S151T variant explains more of the phenotypic variance in Galaxolide intensity rankings (13.27% and 7.54% in discovery and validation cohorts, respectively) than M263T (12.84% and 4.74%). Variant homozygotes (T/T) ranked intensity lower than reference homozygotes (S/S or M/M) by an average of 33.3% and 17.1% (percentage of full scale) for M263T and 34.5% and 31.4% for S151T, for the discovery and validation cohorts, respectively (Fig 3C and 3D).

To search for a mechanistic explanation for the observed associations, we tested high frequency (>5% frequency in validation cohort) OR haplotypes in high LD in this locus (OR4D6, OR5A1, OR5AN1, OR5A2; Fig 3B and S5 Table). None of the ORs in the associated LD-block, or a consensus version of OR4D6 across 10 closely related species [10,32] responded to Galaxolide in our assay.

## OR51B2 variant allele L134F is associated with increased 3M2H intensity

In the discovery study, 3M2H intensity perception was associated with an OR cluster in chromosome band 11p15.4 (Fig 4A and 4B). The peak variant (rs3898917, $p<1.91 \times 10^{-11}$) is in a non-coding region in the LD band including OR51B2, and is in an expression quantitative trait locus (eQTL) affecting OR52A1 [33]. The validation study confirmed that this eQTL is correlated with 3M2H intensity ([low] $p<7.89 \times 10^{-5}$). There are a number of other associated variants in this LD band, but the only variant in the credible set of the fine mapping analysis (S4 Table) was a nonsynonymous missense SNP, rs10837814 (L134F) in OR51B2 (discovery $p<7.83 \times 10^{-10}$, validation $p<9.60 \times 10^{-8}$; Fig 4C and 4D). The meta-analysis confirmed this as the only significant association with 3M2H intensity ([low] $p<8.90 \times 10^{-16}$), and found no further signal for pleasantness (S2 Data).

Given that the evidence from the meta-analysis and fine mapping analysis pointed to OR51B2, we examined the effect size of the L134F on 3M2H intensity ranking. The novel variant explains 4.12% and 9.97% of phenotypic variance in 3M2H intensity rankings in the discovery and validation cohorts, respectively. Variant homozygotes (F/F) ranked intensity higher than reference homozygotes (L/L) by an average of 12.8% in the discovery and 20.8% in the validation (Fig 4C and 4D).

In order to further search for a mechanistic explanation for the observed associations, we used a cell-based assay to measure the response of high frequency (>5% frequency in the validation cohort) OR haplotypes in the associated locus (OR51B2, OR51B4, OR51B5, OR51B6; Fig 4B and S6 Table), as well as in the eQTL-target locus (OR52A1, OR52A4, OR52A5;

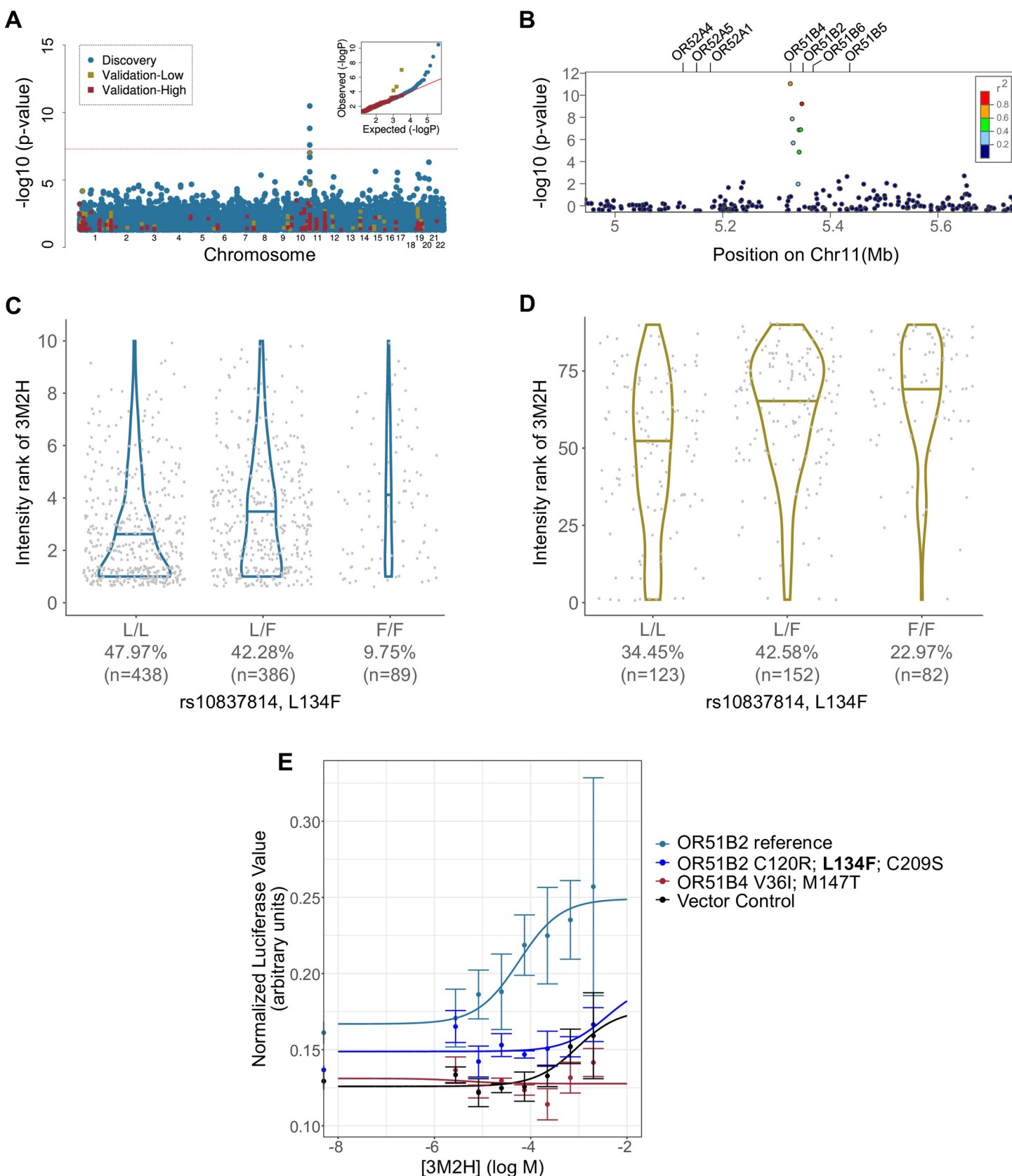

**Fig 4. 3M2H perception is associated with variation in chromosome band 11p15.4 in both cohorts**, as shown by **A)** a Manhattan Plot of associations with the discovery study in blue and validation study in red (high concentration) and yellow (low concentration). The red line indicates the threshold for genome-wide significance (p<5x10⁻⁸). Inset: QQ plots from the discovery (blue) and validation ([high] = red, [low] = yellow) cohorts (Genomic Lambda: discovery study = 1.01; validation study = 0.90) show appropriate control for inflation due to population structure. **B)** Regional plot of discovery study associations indicating both the significance level and the recombination rate at the OR51B2/4 LD-band. The variant L134F (rs10837814; OR51B2) was associated with the perceived intensity of 3M2H in the **C)** discovery and **D)** validation (low concentration) cohorts. The x-axis is ordered left-to-right for increasing number of variant alleles, with population frequency indicated below the genotype. **E)** The OR51B2 reference haplotype responds to 3M2H in a cell-based assay, but the haplotype containing the L134F variant does not. The empty vector control (Rho) does not respond to 3M2H, nor do other receptors in the same LD-band such as OR51B4.

S6 Table), to 3M2H. The OR51B2 reference haplotype responded to 3M2H, but the variant haplotype containing L134F did not (Fig 4E). No other receptors in the OR51B2-associated locus or the eQTL-target locus responded to 3M2H (S2 Fig).

## Replication of previously reported odor phenotype/OR associations

**β-ionone/OR5A1.** We replicated the association between β-ionone intensity perception and the missense SNP rs6591536 (D183N) in OR5A1 in the discovery cohort (p<5.48x10⁻⁴²), the validation cohort ([high] p<7.80x10⁻³⁹; Fig 5A and 5B), and the meta-analysis (p<3.93x10⁻⁷⁵). The pleasantness rank of β-ionone was also associated with D183N in the validation cohort ([high] p<5.53x10⁻¹⁹) and the meta-analysis (p<3.08x10⁻⁹), but not the discovery cohort (p = 0.08). D183N was the top association with β-ionone, and other significant hits were all within the surrounding LD band (S2 Data).

The variant D183N explains 21.6% and 31.9% of phenotypic variance in β-ionone intensity rankings in the discovery and validation cohorts, respectively. Variant homozygotes (N/N) ranked β-ionone intensity lower than reference homozygotes (D/D) by 20% and 38.3% in the discovery and validation cohort, respectively (Fig 5C and 5D).

**Androstenone/OR7D4.** Androstenone perception has been previously associated with the RT/WM haplotype of OR7D4, which consists of two perfectly linked SNPs (rs61729907 (R88W) and rs5020278 (T133M)). This association was directly replicated in the discovery cohort for intensity (p< 9.28x10⁻⁸) and pleasantness (p< 8.64x10⁻⁶) ranking of androstenone. The validation cohort replicated the association for pleasantness (p<0.017), but not intensity (p = 0.10). The meta-analysis found associations with RT/WM and both androstenone phenotypes (intensity p<5.56x10⁻⁸; pleasantness p<5.15x10⁻⁷). The validation cohort also confirmed the published effect of another OR7D4 SNP, rs61732668 (P79L), on androstenone perception (intensity p<3.53x10⁻⁴; pleasantness p<6.88x10⁻³). This SNP was not sequenced or successfully imputed in the discovery cohort and therefore could not be examined in the meta-analysis. In the discovery cohort, there was one novel association that reached genome-wide significance: rs117391865, an intronic SNP nearest the gene SYNE1 (p<1.48x10⁻⁸).

The RT/WM variants in OR7D4 predict 2.5% of phenotypic variance in androstenone intensity rankings in the discovery cohort. WM homozygotes ranked intensity lower than RT homozygotes by an average 20.7% (S3 Fig). The P79L variant in OR7D4 predicts 2.8% of variance in androstenone intensity in the validation cohort. L homozygotes ranked intensity lower than P homozygotes by an average 25.6%.

These findings support previous literature: OR7D4 genetic variation has a consistent effect on androstenone perception, but explains only a small portion of the variance.

**Cis-3-hexen-1-ol/OR2J3.** The association between cis-3-hexen-1-ol intensity perception and rs28757581 (T113A) in OR2J3 was nominally replicated in the discovery cohort (p<0.02), but not in the validation cohort. The meta-analysis results suggest that this is an association at the low (p<0.03), but not high (p<0.08) concentration. In the meta-analysis, there are a number of associations in the LD band surrounding OR2J3 (including OR2W1 and OR2J1) at the

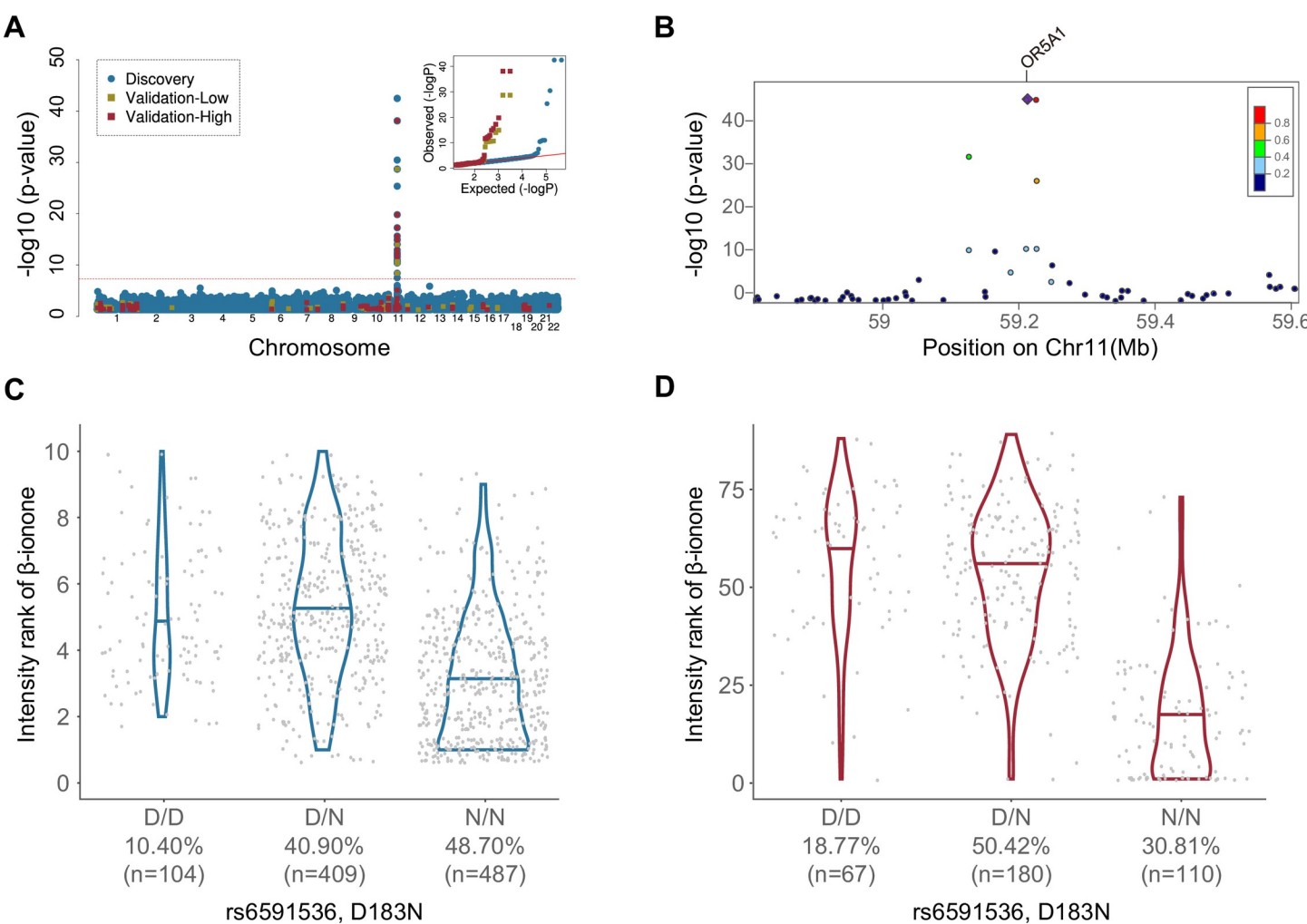

**Fig 5. β-ionone perception is associated with variation in rs6591536 (OR5A1) in both cohorts**, as shown by **A)** a Manhattan Plot of associations with the discovery study in blue and validation study in red (high concentration) and yellow (low concentration). The red line indicates the threshold for genome-wide significance (p<5x10⁻⁸). Inset: QQ plots from the discovery (blue) and validation ([high] = red, [low] = yellow) cohorts (Genomic Lambda: discovery study = 1.00; validation study = 0.94) show appropriate control for inflation due to population structure. **B)** Regional plot of discovery study associations indicating both the significance level and the recombination rate at the OR5A1 LD-band. The previously discovered D183N variant (rs6591536; OR5A1) also changes perceived intensity of β-ionone in our populations: **C)** discovery cohort, **D)** validation cohort (high concentration). The x-axis is ordered left-to-right for increasing number of variant alleles, with population frequency indicated below the genotype.

p<0.05 significance level for pleasantness of cis-3-hexen-1-ol, and the top signal is rs3130765 in OR2J3 (p<1.19x10⁻⁴). No associations for cis-3-hexen-1-ol phenotypes reached genome-wide significance (Fig 2). Previous studies have not consistently demonstrated an association between cis-3-hexen-1-ol and OR2J3. Here we present evidence that some OR in this LD band may be involved in perception of cis-3-hexen-1-ol.

**Caproic acid/OR1A1.** We could not examine the previously published association between OR1A1 and caproic acid in the discovery study as this region was not sequenced or successfully imputed. Although rs17762735 in OR1A1 was associated with intensity in the validation study, the effect of the variant on the phenotype was in the opposite direction from the literature [10]. There were no associations for pleasantness perception of caproic acid with OR1A1. GWAS of caproic acid phenotypes detected a SNP on chromosome 5 associated with pleasantness in discovery (rs56115323, P<1.20×10⁻⁹), however, there is lack of further validation.

There were no associations with aldehyde intensity or pleasantness (Fig 2). No significant signals were discovered for monomolecular aldehydes (decyl aldehyde and galbanum oxathiane) or either fragrance, MixA or MixB. The validation study did not examine these compounds.

## Derived alleles are ancient and associated with reduced odor intensity

Including the two novel SNPs reported in this study, we examined 29 SNPs that have been associated with odor perception in the literature (Table 1). From catalogued data by dbSNP,

**Table 1. Variant age and effect of derived allele on odor perception.** In a literature review, derived alleles corresponded with decreased sensitivity to odor in 21 out of 29 cases, and 11 out of 13 cases with functional validation. All but two variants predate the estimated ages of the East Asian and European population divergences.

| | Gene | Odor | SNP | Derived allele is less sensitive | Frequency of less sensitive allele* | | | Age of derived alleles (years) | Cell assay |
|---|---|---|---|---|---|---|---|---|---|
| | | | | | AFR | EAS | EUR | | |
| 1 | OR5A1[9] | β-ionone | rs6591536 (G>A) | √ | 0.44 | 0.76 | 0.69 | 215,175 | √ |
| 2 | OR7D4[6] | Androstenone | rs5020278 (G>A)[§] | √ | 0.06 | 0.22 | 0.2 | 340,350 | √ |
| 3 | OR7D4[6] | Androstenone | rs61729907 (G>A) | √ | 0.06 | 0.22 | 0.2 | 333,000 | √ |
| 4 | OR4D6 | Galaxolide | rs1453542 (G>C)[†] | √ | 0.05 | 0.17 | 0.28 | 1,240,175 | × |
| 5 | OR4D6 | Galaxolide | rs1453541 (T>C)[†§] | √ | 0.32 | 0.18 | 0.3 | 1,417,950 | × |
| 6 | OR51B2 | 3M2H | rs10837814 (G>A) | × | 0.66 | 0.68 | 0.39 | 1,061,625 | √ |
| 7 | OR2J3[8] | Cis-3hexen-1-ol | rs3749977 (G>A)[†§] | √ | 0.6 | 0.25 | 0.23 | 1,491,850 | √ |
| 8 | OR2J3[8] | Cis-3hexen-1-ol | rs28757581 (A>G) | √ | 0.38 | 0.11 | 0.12 | 853,700 | √ |
| 9 | TAAR5[11,12] | Trimethylamine | rs41286168 (A>G) | √ | 0 | 0.001 | 0.01 | 11,625 | √ |
| 10 | OR6C70[11] | Licorice | rs60683621 (C>G)[§] | × | 0.79 | 0.51 | 0.81 | 113,475 | × |
| 11 | HBG2[11] | Cinnamon | rs317787 (T>C)[§] | √ | 0.57 | 0.51 | 0.68 | 1,471,450 | × |
| 12 | OR10J5[10] | Bourgenol | rs35393723 (G>A)[§] | √ | 0.03 | 0.02 | 0.19 | 84,300 | √ |
| 13 | OR11A1[10] | 2-ethyl fenchone/ Fenchol | rs9257857 (C>T)[§] | √ | 0.05 | 0.08 | 0.15 | 117,200 | √ |
| 14 | OR1C1[10] | Linalool | rs116453035 (G>A) | √ | 0.07 | 0 | 0 | 41,550 | √ |
| 15 | OR2W1[10] | caproic acid | rs35771565 (C>T) | × | 0.76 | 0.61 | 0.83 | 113,900 | √ |
| 16 | OR2A5: 2kb Upstream Variant[10] | citronella | rs869068021 (AC/A) deletion | × | / | | | / | × |
| 17 | OR6Y1[10] | diacetyl | rs41273491 (C>T)[§] | √ | 0.05 | 0.38 | 0.24 | 318,550 | √ |
| 18 | OR6Y1[10] | diacetyl | rs16840314 (G>A)[§] | √ | 0.03 | 0.38 | 0.24 | 112,075 | √ |
| 19 | OR2A25[10] | citronella | rs59319753 (G>C)[§] | × | 0.22 | 0.63 | 0.21 | 558,975 | × |
| 20 | OR10G4[10] | guaiacol | rs4936880 (A>G)[†] | √ | 0.47 | 0.36 | 0.44 | 1,211,375 | × |
| 21 | OR10G4[10] | guaiacol | rs4936881 (A>C)[†] | √ | 0.47 | 0.36 | 0.44 | 1,268,275 | × |
| 22 | OR10Z1[10] | diacetyl | rs857685 (A>C) | √ | 0.07 | 0.5 | 0.26 | 175,750 | × |
| 23 | OR10C1[10] | 2-ethylfenchol | rs2074466 (C>A)[§] | √ | 0.04 | 0.25 | 0.14 | 114,275 | × |
| 24 | OR6B2[10] | isobutyraldehyde | rs10187574 (A>G) | √ | 0.16 | 0.46 | 0.35 | 459,925 | × |
| 25 | OR5F1: 500b Downstream Variant[10] | orange | chr11: 55761081 (TA>T) deletion | √ | / | | | / | × |
| 26 | OR8A1: 3 Prime UTR Variant[10] | paraffin oil | rs7931189 (A>T)[†] | × | 0.16 | 0 | 0 | 413,450 | × |
| 27 | OR6Y1[10] | 2-butanone | rs41273491 (C>T)[§] | × | 0.05 | 0.38 | 0.24 | 318,550 | × |
| 28 | OR10G4[10] | guaiacol | rs4936882 (T>G)[†§] | √ | 0.52 | 0.72 | 0.73 | 1,274,850 | × |
| 29 | OR2M7: 9kb 5' of OR2M7[13] | Asparagus Urine | rs4481887 (G>A)[†] | × | 0.12 | 0.14 | 0.26 | 1,225,700 | × |

*AFR, African; EAS, East Asian; EUR, European. Frequency data were calculated from 1000 Genome Project.

[†]The mutations existed in archaic humans.

[§] The mutations existed in other primates.

we extracted information about the evolutionary status of each polymorphism, including the estimated age of when the mutation occurred, and which allele was derived (mutated) versus ancestral. In 24 of the 29 SNPs, the age of the derived allele was more than a hundred thousand years old (112,075–1,491,850 years), predating the population divergence times between East Asians and Europeans (~55,000 years ago for East Asians and ~41,000 years ago for Europeans) [34]. Several SNPs existed in archaic humans and other non-human primates (Table 1). Based on the Composite of Multiple Signals (CMS) score, there was no sign of natural selection for any of the 29 SNPs (S4 Fig).

In 21 out of 29 examined cases the derived allele was less sensitive to odors (72.4%; $p < 0.01$). 13 of these 29 SNPs have been functionally validated by cell assay. Of these 13, there were 11 cases where the derived allele associated with decreased odor sensitivity (84.6%; $p < 0.01$).

## Discussion

We conducted a genome-wide association study using ten odors and found novel associations for OR4D6 with the musk odor Galaxolide, and OR51B2 with 3M2H. In addition, we replicated previous associations between OR5A1/β-ionone, OR7D4/androstenone, and OR2J3/cis-3-hexen-1-ol. Furthermore, we have shown that these genotype/phenotype associations are stable across populations and robust to differences in methods, including odor concentration and delivery method. Previous genotype/phenotype studies have tended to focus on variation in olfactory receptors, however differences in olfactory perception could be driven by genetic variation in other proteins involved in odor signal transduction, such as olfactory axon guidance molecules, odor-modifying enzymes, or odor transport proteins. Despite our genome-wide search, the peak associations were largely located within olfactory receptor loci, suggesting that differences in olfactory perception caused by genetic factors are frequently driven by changes in the receptors. It should be noted that the odor perception assessment methods are different across cohorts, which will inevitably lead to some discrepant results across cohorts.

### OR4D6 variation drives differences in perception of Galaxolide, but multiple receptors are involved in musk perception

Musks are a chemically diverse set of compounds that are defined by their common perceptual quality; however variation in perception of intensity of different musk odors across individuals [16,17] suggests several receptors or groups of receptors may have a contributing role. The musk family, therefore, provides us with an opportunity to study the convergence of perceptual features of odors through differential receptor activation in the olfactory code. Prior to our study, there were four human olfactory receptors that responded to musks in cell culture [18,30,31], but their influence on olfactory perception is unknown. Here we identified a fifth musk receptor, OR4D6, where genetic variation associated with differences in perception of the polycyclic musk, Galaxolide. This is the first behavioral evidence that any human olfactory receptor plays a role in musk perception.

Other receptors that may be involved in musk perception have shown specificity for a particular musk or musk chemical family. Mice with a genetically deleted Olfr1440 (MOR215-1) were unable to find muscone in an odor-finding task [30], suggesting that the receptor is necessary for detection of the polycyclic musk muscone. The human ortholog of Olfr1440, OR5AN1, has relatively high affinity for several macrocyclic and nitro musk compounds in a heterologous cell-based assay [18,30]. Screening with this cell-based assay uncovered two other putative human musk receptors, OR1A1 and OR2J3 [30], which respond only to nitro musks, but not Galaxolide or other polycyclic musks. There is also recent evidence for a

broadly tuned musk receptor, OR5A2, which is activated by musks from all four tested structural classes in vitro [31]. Together, the existence of musk-specific or musk-family specific, as well as broadly tuned musk receptors suggests that musks activate separate, but potentially overlapping, sets of receptors.

Here, we have identified the first case where genetic variation in a receptor is associated with musk perception in humans. Although OR4D6 is the top association with the Galaxolide intensity phenotype, it is in high linkage disequilibrium with two previously identified musk receptors: OR5AN1, and OR5A2. Due to solubility issues with Galaxolide in our cell-based assay, we were unable to provide functional evidence for OR4D6; however, several pieces of evidence support the idea that genetic changes in OR4D6 are driving the phenotypic difference in Galaxolide intensity: 1. In both cohorts OR4D6 is a stronger predictor of Galaxolide intensity than OR5A2 and OR5AN1, which are not significantly associated with the phenotype after controlling for the top variant, and 2. With few exceptions, participants homozygous for the OR4D6 variant are unable to smell Galaxolide (S5 Fig).

An interesting finding is that Galaxolide-associated rs1453541 and Beta-ionone-associated rs6591536 are in LD ($R^2 = 0.72$) as they are only 14kb apart. The SNP rs6591536 was functionally validated to be the causal variant affecting the perception of Beta-ionone [9]. However, it is unlikely to be the causal variant affecting the perception of Galaxolide for the following reasons: 1) fine-mapping using PAINTOR found the 99% credible set only contain SNPs rs1453541 and rs1453542, with the posterior probability of 58% and 42%, respectively. The probability of rs6591536 being the causal variant is close to zero. 2) We have performed cell-based assays using a number of SNPs, including rs6591536, but found no significant response (S5 Table). 3) If rs6591536 was indeed the causal SNP for the perception of Galaxolide, one would expect the two phenotypes (perception of Galaxolide and Beta-ionone) should be correlated to some extent, but this is not the case ($\rho = -0.01$, Pearson's correlation).

OR4D6 is a strong candidate for the mechanism underlying specific anosmia to Galaxolide, suggesting that it is possible for a single receptor to represent the musk percept. We do not know if OR4D6 contributes to perception of other musk compounds, but given the in vitro evidence for other musk receptors and behavioral data that suggest those with Galaxolide anosmia are still able to smell other musk compounds [16,17], it is unlikely to be solely responsible for the perception of the musk quality percept. OR4D6, OR5AN1, and OR5A2 are prime targets for future work on musks, which can lead more broadly to understanding how activation of different combinations of receptors results in highly similar percepts.

## OR51B2 variation drives differences in the perception of human body odor component 3M2H

Trans-3-methyl-2-hexenoic acid (3M2H) has been described as the 'impact odor' for body odor arising from the underarms, meaning that it is a highly abundant volatile compound and its quality as a monomolecular odorant is the same as the characteristic quality of body odor [23]. Specific anosmia to 3M2H has been reported in several studies with rates ranging from 21–25% of the population [23,25,26]. Based on its key role in body odor character, it is likely that anosmia to 3M2H alters body odor perception, although it does not eliminate the ability to smell body odor, as there are other reported volatile compounds present in underarm odor [23,26,35].

Here we found that OR51B2 was associated with 3M2H intensity, and responded to 3M2H in a functional assay. Although the OR51B2 haplotype containing the L134F variant is associated with a higher intensity of 3M2H, it did not respond to 3M2H in a functional assay. This is surprising, as the more functional variant is typically associated with a higher perceived

intensity. One possibility is that the *in vitro* assay does not perfectly replicate the environment of the OSN—the assay has previously failed to provide direct support for genetic associations [10]. Despite this, the consistent functional response of the reference OR51B2 to 3M2H alongside the validated genetic evidence still strongly suggests that OR51B2 drives differences in perception of 3M2H intensity. This finding suggests that OR51B2 genotype will impact the perception of body odor. OR51B2 could be a target for future studies interested in malodor blocking, or discovering the mechanisms underlying social communication from body odor. It is worth noting that the distribution of 3M2H intensity was different among the discovery and validation cohorts, which may be contributed by many factors, such as different concentrations of odors, different methods for delivering the odors, and different allele frequency of OR51B2.

### Associations between OR5A1/β-ionone, OR7D4/androstenone, and OR2J3/cis-3-hexen-1-ol are replicated in an East Asian population

In the East Asian discovery cohort, we replicated previous phenotype/genotype associations between OR5A1/β-ionone, OR7D4/androstenone, and OR2J3/cis-3-hexen-1-ol, but failed to replicate the OR1A1/caproic acid association. The N183D OR5A1 variant has now been associated with decreased intensity perception of β-ionone in several studies [9,36], as well as verified in a cell-based assay [9]. In both the discovery and the validation cohorts, the β-ionone intensity phenotype had the highest overall effect size, showing this association is not only robust to differences in methods, but has also been replicated across multiple populations.

In the discovery cohort, we replicate the association between RT/WM haplotypes of OR7D4 and androstenone perception [6], that has been replicated in two other populations [37,38]. Although the validation study did not replicate the association with androstenone intensity and RT/WM, it did replicate the association with androstenone pleasantness, as well as the previously discovered association between P79L in OR7D4 and androstenone perception [6]. The lack of signal for intensity perception and RT/WM in the validation study could be due to differences in odor delivery method or concentration of the odor, but is more likely due to lack of power in the smaller validation cohort given that this association was replicated in the meta-analysis. Overall, the evidence here continues to support the role of OR7D4 in androstenone perception.

Here we nominally replicated the association between cis-3-hexen-1-ol and OR2J3[8]. The smaller signal here is not surprising, as this association has failed to replicate in two other studies [10,36]. The original discovery of this association measured the detection threshold of cis-3-hexen-1-ol, while the two studies that failed to replicate the original association measured intensity rankings. Since the set of receptors that associate with variation in olfactory perception differs across concentrations [10], this could explain why the cis-3-hexen-1-ol/OR2J3 association failed to replicate previously and only has a small signal here. It is important to acknowledge that for all of our genetic associations, our predictive power is limited by the quality of our participant data, as measured by within-subject test-retest correlation. While it may be possible in future studies to improve test-retest by conducting additional training of participants, the overall test-retest correlation in our study is on the high end of the subject reliability range compared to the larger field of olfactory psychophysics. This may be influenced, in part, by our study design, which focused on a set of odors known to have high phenotypic variability.

A previous study found an association between OR1A1 and caproic acid [10]. This failed to replicate in the validation study, and had no direct replication provided from the discovery study. The discovery study did have data on variants in the locus surrounding OR1A1 that had

an association signal with caproic acid p<0.05. The role of OR1A1, or perhaps another OR in this region, in the perception of caproic acid is still unclear.

The majority of previous odor association studies have been conducted in heterogeneous and majority-European populations, leaving a fundamental gap in knowledge of the wider relevance of these associations to different populations, such as the East Asian population examined here. We found that associations tend to replicate across different populations. Summarizing the variants that alter odor perception, including those reported in past literatures, our analysis of evolutionary age found 25 of 27 variants predated the population divergence between East Asians and Europeans (~55,000 years ago for East Asians and ~41,000 years ago for Europeans) [34]. These variants are generally present in both populations at relatively high frequencies. On the other hand, the two more recently derived variants (S95P in TAAR5 and A67V in OR1C1) have very low minor allele frequencies in both East Asian and European populations, suggesting population specific variants that alter odor perception are rare.

## Variation in the perception of aldehydes does not associate with olfactory receptors

A study in a large non-homogenous population from New York, NY, tested perceptual differences of 15 fragrances between self-reported demographic groups. Of all the odors tested, the largest difference found was in aldehydes, and all three tested aldehydes (decyl aldehyde, nonyl aldehyde, and undecanal) were significantly different, such that the Asian population perceived aldehydes to be more intense than the Caucasian population [28]. The follow-up study pursuing the genetic underpinnings of these differences did not identify any associated ORs [10], and even here, in a larger cohort with a genome-wide search, there were no associations.

The lack of genetic evidence here may also be due to the involvement of multiple ORs, reducing our power to detect specific associations; or the perceptual variation may be due to cultural, social, or other factors that are not genetic in nature. The more recent evolutionary age of population-specific variants may play a role, as this these types of odor analyses have discovered mostly ancient variants, with only two odor-associated SNPs that appeared after the East Asian and European population divergences. This suggests that increasing population size, regardless of diversity, may be necessary to discover more recently derived SNPs with lower minor allele frequency.

## Degeneration of olfactory receptor gene repertoires in primates

Compared to many non-primate mammalian species, primates have fewer intact olfactory receptor genes both in absolute number and by percentage [39]. While previous analyses have been restricted to pseudogenes, recent analyses of the functional consequences of missense mutations allow for a more detailed examination. We found that in 72% of reported OR gene/olfactory phenotype associations reported in the literature (85% with functional validation), derived alleles predicted lower perceived intensity than ancestral alleles. While this study was not designed to directly address this hypothesis and may suffer from selection bias, these data support the hypothesis that the primate olfactory gene repertoire has degenerated over time. The functional implications of this degeneration remain unclear [40,41].

## Large genetic databases can be used to understand OR function, a proxy for general protein function

In the discovery study, we may have the benefit of measuring olfactory phenotypes in a large, homogenous cohort (Fig 1) where genome-wide genotyping had already been conducted,

giving us the statistical power of a large population without the time or expense. In this study, the novel signals do not have much population differences in MAF or effect size (Table 1 and Figs 3 and 4), suggesting that the large sample size rather than its genetic similarity might be the more important reason behind the findings. Given the increasing number of open databases of sequencing data, this method is becoming a more reasonable possibility for easily testing genotype/phenotype associations.

Olfaction is an excellent use of this new resource because of the ease of understanding the functional output of genetic variation in the protein. The human olfactory system has both robust assays to test the behavioral output of these proteins (psychophysics/rating odors) [5,6,10] and an established method for directly testing protein function in cells (heterologous cell-based assay) [42,43]. Genetic variation provides a strong tool for exploring olfactory coding and sheds light on how complex systems integrate information from variable sensors.

## Materials and methods

### Ethics statement

The discovery cohort was conducted under approval of the Ethics Committee of Chinese Academy of Sciences (Shanghai, China). The validation cohort was conducted under approval from the IRB at Rockefeller University (New York, NY). Written formal consent was obtained in both cohorts.

### Study cohorts and participants

The discovery cohort comprised 1000 participants between the ages of 18 and 55, from a Han Chinese population collected in Tangshan, China. The validation cohort comprised 364 participants between the ages 18–50, from a diverse population collected in New York, New York, USA.

Both cohorts excluded participants with medical conditions that affect the sense of smell, specifically: smoking, recreational drug use, brain surgery or head trauma that required hospitalization, chronic nasal issues (allergic, tumoral, infectious or inflammatory disease), history of endoscopic nasal or sinus surgery, any neurodegenerative disease, any upper respiratory infection that altered the sense of smell and/or taste for more than 1 month, cervicalgia or other neck diseases, history of radiation or chemotherapy, alcoholism, current sinus or upper respiratory infection, seasonal allergic rhinitis or acute rhinosinusitis, and use of medications that interfere with the sense of smell.

### Odor delivery

Discovery cohort participants were tested using felt-tip pens (100.2 mm length, diameter 7.7 mm; ETRA, Königsbach-Stein, Germany) containing an absorbent material loaded with 1 mL of liquid odor. Each pen was used for no more than 50 participants before being discarded. After preparation, individual sticks were used within 2 months.

Validation cohort participants smelled 20 ml amber glass vials filled with 1mL of odor. The vials were presented in a double-blind manner, labeled only with barcodes, to prevent experimenter bias.

### Phenotyping

Discovery cohort participants smelled 11 odors (10 unique and one repeat) and verbally rated the intensity and pleasantness on a 100-point scale. For each odor, a unique set of 100 participants rated the stimulus twice so we could measure the test-retest reliability (S6 Fig).

Validation cohort participants also smelled each stimulus and rated intensity and pleasantness on a 100-point computerized sliding scale. The participants smelled 46 odors at one concentration, 26 odors at two concentrations, three odors at five concentrations, and three solvents for a total of 90 stimuli, ten of which are reported here (six odors: four at low and high concentration and two at high concentration only). Participants smelled stimuli in the same order to facilitate comparisons across participants, and every stimulus was presented twice.

In both cohorts, in order to normalize for scale usage across raters, intensity and pleasantness ratings for each participant were ranked from 1 to 10, or 1 to 90, such that the odorant with the lowest rated intensity was ranked at 1, and the odorant with the highest intensity was ranked 10 or 90, depending on the total number of stimuli (S1 Fig) [6]. The change in ranking metric was calculated as a percentage of the number of ranks changed over the total number of ranks in the scale (10 or 90), in order to directly compare changes between the cohorts.

To measure the within-subject reliability of ratings, we calculated the Pearson's correlation between duplicate stimuli. Due to methodological constraints of collecting duplicate participant ratings in the discovery cohort, we conducted this analysis with the raw participant data, prior to ranking, In the validation cohort, 4 participants were removed due to poor performance (test-retest correlation < = 0). Individual performance could not be examined in the discovery cohort, as only one stimulus was repeated for each participant.

## Odor concentration and preparation

The discovery cohort included 8 monomolecular odorants (androstenone, β-ionone, caproic acid, cis-3-hexen-1-ol, Galaxolide, trans-3-methyl-2-hexenoic acid (3M2H), decyl aldehyde, and galbanum oxathiane) and 2 odor mixtures (MixA and MixB) all prepared by Unilever. We used isointense concentrations of the ten odors that were diluted in either propylene glycol or MCT (medium chain triglycerides) (Tables 2 and S7). To determine the concentration, we utilized a panel of 14 experts from a sensory flavor panel trained by Unilever. These panelists

**Table 2. Concentrations of Odors from the Discovery and Validation Studies.**

| Odor [Alternate Name] | Discovery Cohort | | Validation Cohort | |
|---|---|---|---|---|
| | Dilution | Solvent | Dilution (high/low) | Solvent |
| β-ionone [3E-4-(2,6,6-Trimethylcyclohex-1-en-1-yl)but-3-en-2-one] | 50X | Propylene Glycol | 1/10,000 1/400,000 | Paraffin Oil |
| 3M2H [trans-3-methyl-2-hexenoic acid] | 0.1 g/mL | Propylene Glycol | 1/100 1/20,000 | Paraffin Oil |
| Galaxolide [1,3,4,6,7,8-hexahydro-4,6,6,7,8,8-hexamethylcyclopenta[g]-2-benzopyran] | 2X (50% in diethylphthalate) | MCT (medium chain triglycerides) | 1/10 1/1000 weight/Volume | Paraffin Oil |
| Cis-3-hexenol | 200X | Propylene Glycol | 1/100,000 1/250,000 | Paraffin Oil |
| Decylaldehyde [decanal] | 1000X | Propylene Glycol | n/a | n/a |
| Androstenone [5α-androst-16-en-one] | 1.42 mg/mL | Propylene Glycol | 1/1000 weight/Volume | Propylene Glycol |
| Caproic acid [hexanoic acid] | 500X | Propylene Glycol | 1/1,000,000 | Paraffin Oil |
| galbanum oxathiane [(2R,4S)-2-methyl-4-propyl-1,3-oxathiane] | 5000X | Propylene Glycol | n/a | n/a |
| MixB in diethylphthalate | 1000X | Propylene Glycol | n/a | n/a |
| MixA | 400X | Propylene Glycol | n/a | n/a |

rated intensity of ten odorants at three different concentrations (except 3M2H and androstenone which were rated at 2 concentrations) that were pre-selected to cover a range from weak to strong. Panelists rated intensity on a scale from 0–15, using a range of concentrations of citric acid for reference. Ratings were significantly different between all concentrations of odors, except for androstenone, for which we chose the higher concentration. For each odor, we chose the concentration that was closest to an intensity of 7, with the exception of two odors (caproic acid and MixA) for which an original concentration did not result in a rating near 7. For these odors, we extrapolated the concentration that would result in an intensity rating of 7 from the other intensity ratings.

The validation cohort includes data from the following six odors: androstenone, β-ionone, cis-3-hexen-1-ol, caproic acid, Galaxolide, and 3M2H. The aldehydes and fragrances were not measured in the validation study. High and low concentrations of odors were intensity-matched to 1/1,000 and 1/10,000 dilutions of 1-butanol, as determined by rankings from a panel of 13 individuals. Odors were presented at both concentrations, except for androstenone and caproic acid, which were given at concentrations based on previous studies [6,10]. Odors were diluted in paraffin oil or propylene glycol (Table 2).

Due to different delivery methods in each cohort, the concentrations of these six compounds cannot be directly compared to the concentrations in the discovery study [44,45].

## Genotyping

Discovery Cohort: Genomic DNA was extracted from blood samples using the MagPure Blood DNA KF Kit. All samples were genotyped using the Illumina Infinium Global Screening Array that analyzes over 710,000 SNPs. It is a fully custom array designed by WeGene (https://www.wegene.com/).

Validation Cohort: Genomic DNA (gDNA) was extracted from saliva samples using the Oragene Discover 2mL kit and protocol. Library prep (using Agilent SureSelect XT2 kit) and targeted sequencing were performed by CAG sequencing core (Children's Hospital of Philadelphia Research Institute, Philadelphia PA). Custom Agilent SureSelect targets were designed (eLID# 3028991) for 418 ORs and 290 olfactory-related genes, including other odorant receptors (i.e. TAARs, MS4A) and related enzymes (i.e. CYP). The Illumina HiSeq platform was used to perform paired-end sequencing with a read length of 2x125 basepairs on 364 participants.

## Variant calling and quality filtering

Discovery Cohort: Sequences were aligned to genome build GRCh37/hg19 and genotypes were called using Genome Studio v2.0[46]. To control for genotype quality, we implemented exclusion criteria using PLINK v1.90b6.9 [47]. No people were removed due to >5% missing data or failure of X-chromosome gender concordance check. We excluded SNPs that had >2% missing data (14,385 variants removed), a minor allele frequency (MAF) <1% (251,918 variants removed), or a deviation from Hardy-Weinberg (HW) equilibrium ($p < 1 \times 10^{-5}$) [48] (1,149 variants removed), leaving 433,485 SNPs from 1000 individuals for genome wide association analysis. SNP phasing was performed with Eagle v2.4[49] using 1000G Phase 3 V5 (GRCh37/hg19) EAS as the reference panel [29]. We conducted imputation on the 433,485 phased SNPs using Minimac4, and obtained a total of 45,843,286 variants. We then re-ran genotype quality control steps and filtered out 54 variants missing >2% genotype data, 27,361 variants with a deviation from HW equilibrium ($p < 1 \times 10^{-5}$) [48], and 37,772,956 variants due to MAF threshold (MAF>0.01), leaving 8,042,915 variants for association analysis.

Validation Cohort: Genotypes were called using a pipeline that follows recommended 'best practices' by the Broad Institute [50,51], and as previously reported [10]. Sequences were aligned to GRCh37/hg19 genome build using BWA [52], and alignment, genotype quality and variant calling steps were performed using Picard Tools [53,54]. SNP phasing was performed with SHAPEIT V2.r900 [55], and OR haplotypes were assembled using a custom R script. Of the original 18,611 variants called, quality control measures filtered out 1,488 SNPs that were missing genotype data at a frequency >5% or deviated from Hardy-Weinberg equilibrium ($p<1x10^{-5}$) [48]. An additional 14,078 variants were removed due to minor allele frequency (MAF>0.05), leaving 3,045 variants for association analysis. We adjusted for a more stringent MAF cutoff in the validation cohort, as the sample size of the validation cohort is relatively small compared to the discovery cohort. Three individuals were excluded due to > 5% missing data, leaving 357 participants remaining for genotype/phenotype analysis. For one region of the genome (chromosome band 11p15.4) the discovery study found significant association in a non-coding region. This region was not sequenced in the validation study, which focused on open reading frame variants, so we imputed 147,613 SNPs in this region (11:79438 to 11:249222325, hg19).

## Population structure analysis

We combined the discovery and validation datasets in order to visualize and quantify differences in the two study populations. We performed principal component analysis (PCA) using 990 linkage disequilibrium-pruned (r2<0.2) SNPs from the combined discovery (n = 1000), validation (n = 357), and 1000 Genomes Project (phase 3, 271 participants: 97 CHB, 86 CEU, and 88 YRI) datasets [29]. We calculated centroids for each population using the first two eigenvectors. The distances between populations were measured by Euclidian distance of the centroids. The distances within a population were measured by averaging the Euclidian distance between each point (participant) and the centroid in the population.

## Association analysis

Discovery Cohort: To control for population stratification, we identified the top 10 genetic eigenvectors to use as covariates by performing PCA on 143,988 LD-pruned (r2<0.2) SNPs from the 1000 participants of the discovery cohort using Plink v1.90b6.9 [47,56].

Using PLINK (v1.90b6.9) [47], we performed genome-wide association analyses of 8,042,915 SNPs against 20 ranked phenotypes (intensity and pleasantness of 10 odors) under an additive linear model including age, sex, and the top ten genetic eigenvectors as covariates (S7 Fig). Associations were significant if they passed the conventional genome-wide significance threshold ($p<5x10^{-8}$) [57,58]. For loci of interest, we calculated linkage disequilibrium using LocusZoom[59] using the genome build from hg19/1000 genomes Nov 2014 ANS. We estimated the heritability of each perception phenotype explained by LD-pruned SNP set (143,988 SNPs with r2<0.2) using GCTA software (v1.93.0 beta) [60,61].

Validation Cohort: To determine the top 10 genetic eigenvectors [56,62] for the validation study, we conducted PCA on 10,927 LD-pruned (r2<0.05) SNPs with <5% missing genotypes and in HW equilibrium (p<1e-5) (but without excluding for MAF) from 361 people (including participants later excluded for poor phenotype data) using the R/Bioconductor package SNPRelate [63].

We performed genetic association analysis using PLINK (v1.90b5) [47] to test additive linear models for the 3,045 SNPs from quality control steps and the 78,904 SNPs from the imputed region against each of the 20 phenotypes of interest (intensity and pleasantness of six odors at one or two concentration each; see Table 2) with the top ten genetic eigenvectors as

covariates. For significant loci from the discovery study we set alpha = 0.05. For these loci of interest, we calculated linkage disequilibrium using LocusZoom [59] with the genome build from hg19/1000 genomes Nov 2014 EUR.

Combined Cohorts: We conducted a meta-analysis of the discovery and validation cohorts with METAL [64], which combines weighted p-values, weighted by sample size, across studies while taking into account and direction of effect.

### Fine mapping analysis

We conducted the fine-mapping analysis by leveraging functional annotation data (GenCode. exon.hg19) and LD information in the discovery and replication cohorts [65]. We assumed a single causal variant at each locus, examined the SNPs within 200kb upstream and downstream of the top variant, and calculated the posterior probabilities using PAINTOR to determine the 99% credible set. The 99% credible set was constructed by 1) ranking all variants according to their Bayes factor, and 2) including ranked variants until their cumulative posterior probability of representing the causal variant at the given locus $\geq$0.99.

### Olfactory receptor cloning and haplotypes

To determine functional consequences for the identified SNPs in the olfactory receptors (ORs) and nearby receptors in high linkage disequilibrium, we tested activation of specific haplotypes of the associated ORs, as well as nearby ORs in the same LD-band. We have a large library of variant and reference haplotypes of ORs that we can use for testing differential response of receptor variants in the cell-based Luciferase assay. To supplement our library, we ordered and subcloned the variant haplotype of OR51B2 containing L134F, and the OR4D6 consensus sequence, both into the vector pCI-RHO (GenScript). pCI-Rho (Promega) contains the first 20 amino acids of human rhodopsin [66]. Using a consensus version of a receptor can improve surface expression in a heterologous cell-based assay where the original receptor is not expressed [10,32].

We created a consensus sequence for OR4D6 using orthologs found in Homo sapiens, Gorilla gorilla, Pan paniscus, Pan troglodytes, Pongo abelii, Macaca mulatta, Mandrillus leucophaeus, Callithrix jacchus, Microcebus murinus, Rattus norvegicus, and Mus musculus. We aligned the orthologs using the online version of MAFFT version 7 [67], and determined the most common amino acid at each position for the open reading frame of OR4D6. The consensus amino acid sequence was printed by GenScript and subcloned into the pCI-Rho vector (Promega).

### Luciferase assay

We used a heterologous cell-based assay to determine the functional changes caused by different OR haplotypes for our two novel associations, as has been previously described [10,42,43].

Transfection: Using the Dual-Glo Luciferase Assay System (Promega). We transfected Hana3A cells with our OR of interest, firefly luciferase driven by a cyclic AMP response element (CRE) promoter, and Renilla luciferase driven by a constitutively active SV40 promoter, RTP1S63, and M3-R [68].

Stimulation: Approximately one day after transfection, we stimulated cells by adding the odor in a 3-fold dilution series in CD293. Each concentration was run in triplicate, including the empty vector negative control. Stock odors were kept at 1M in DMSO and diluted in CD293 to the highest applied concentration of 1mM. Four hours after adding odor to cells, we read the luminescence output using a Synergy 2 plate reader (BioTek). Luciferase values were

normalized by Renilla luciferase to control for transfection efficiency and cell death, and then averaged across the triplicate readings.

Analysis: The normalized luciferase values were fit to a three-parameter sigmoidal curve with a fixed slope (slope = 1). We considered a receptor to be activated by an odorant if the response passed three tests: 1) the standard error of the logEC50 was less than one log unit, 2) The 90% confidence intervals for the top and bottom parameters of the curve did not overlap, and 3) The dose response curve from the OR-transfected cells was significantly different from the negative control (empty vector), as calculated by the extra sum-of-squares test. Data analysis was performed using GraphPad Prism (Version 8).

## Evolutionary analysis

We accessed the dbSNP database (https://www.ncbi.nlm.nih.gov/snp/) to determine the derived and ancestral alleles for our two novel SNP associations and 29 SNPs with previously reported odor phenotype associations [5–11,13,69]. To our knowledge, this included all previously published associations between a SNP and an olfactory phenotype, exclusive of haplotype associations where direction of effect from individual SNPs could not be determined. We estimated the age of derived alleles using a Genealogical Estimation of Variant Age (GEVA) model (https://human.genome.dating/) [70]. We checked if these mutations existed in archaic humans (i.e. Neandertal and Denisova) or in other primates using publicly available sequences (https://genome.ucsc.edu/Neandertal/,http://cdna.eva.mpg.de/denisova/) [71] and UCSC database (https://genome.ucsc.edu/). We also tested whether these SNPs were under positive selection using the Composite of Multiple Signals (CMS) method [72]. This method generates a composite score based on three distinct signatures of selection: long-range haplotypes, differentiated alleles, and high frequency derived alleles. To examine the relationship between derived alleles and a decrease in odor intensity perception, we performed a one sided two-proportions z-test (R version 6.3.1).

## Supporting information

**S1 Data. Significant Discovery Cohort Associations (p< 5x10-8).** Matching validation data is included for regions where data is present.
(CSV)

**S2 Data. Meta-Analysis Results.** Shown are all significant (p<5x10-8) associations for the meta-analysis of discovery and validation cohorts at both concentrations of odor (n = 1357).
(CSV)

**S3 Data. Meta-Analysis Results for Replicated Odors.** Shown are all significant (p<0.05) associations for the meta-analysis of discovery and validation cohorts (n = 1357) for associations replicated from the literature. We examined regions within +-200k bp of the original SNP association.
(CSV)

**S1 Table. Heritability of ranked intensity and ranked pleasantness of 10 odors estimated by GCTA software using LD-pruned variants (143,988 SNPs with r2<0.2) from the discovery study.**
(XLSX)

**S2 Table. Frequency of the two SNPs in OR4D6, rs1453541 (M263T) and rs1453542 (S151T) in discovery and validation cohorts.** Haplotypes with the T variant from S151T

always have the T variant from M263T.
(XLSX)

**S3 Table. The associations between Galaxolide and SNPs of other reported musk-related ORs in the discovery cohort (n = 1000) before controlling for the top associated variants (SNPs in OR4D6).** OR5AN1 and OR5A2 are in the same LD-band as OR4D6 (see main Fig 2.), meaning variants in these ORs are more likely to be inherited with the SNPs from OR4D6. After performing an additional analysis controlling for the top associated SNPs in OR4D6 (p-value after controlling for top SNP), we found no additional significant signal.
(XLSX)

**S4 Table. SNPs in the 99% credible set from the fine mapping analysis.** For each odor intensity phenotype, we examined SNPs 200kb upstream and downstream from the top associated SNP. We used PAINTOR to calculate posterior probability based on functional annotation linkage disequilibrium. In the case of two highly linked SNPs, such as with OR5A1 and OR4D6, the posterior probabilities sum to 99%.
(XLSX)

**S5 Table. Olfactory receptor haplotypes (hg19) tested in the cell-based assay for activation by Galaxolide (OR4D6 Cluster).** The bolded variants are the SNPs associated with change in Galaxolide perception. OR4D6 2 is a consensus version of OR4D6 across 10 closely related species.
(XLSX)

**S6 Table. Olfactory receptor haplotypes tested in cell assay for activation by 3M2H (OR51B2/4 Cluster).** The bolded variants are the SNPs associated with change in 3M2H perception.
(XLSX)

**S7 Table. Odor Purity and Origin.**
(XLSX)

**S1 Fig.** Distribution of ranked intensity (A) and pleasantness (B) ratings for odors in the discovery (blue) and replication (red) studies. A grey box indicates the phenotype was not tested.
(TIF)

**S2 Fig. Cell-based assay results for 3M2H against other receptors in the A) OR51B2 and B) OR52A1 clusters.** No receptors responded significantly above the vector control (Rho). Luciferase values were normalized by RL readings and then baselined to zero by subtracting the response of the no-odor control.
(TIF)

**S3 Fig. Intensity perception of androstenone is associated with RT/WM haplotype of OR7D4 in the discovery cohort.**
(TIF)

**S4 Fig. Results for natural selection on candidate OR gene regions (±2kb).** CMS scores are plotted against chromosome position in CEU, CHB+JPT, and YRI populations, shown in blue, gray, and green, respectively. The red dotted line represents the significance threshold (top 0.1% CMS score: 4.791). No enrichment for high CMS scores (top 0.1%) is found within the genes, indicating the examined SNPs are not subject to natural selection.
(TIF)

**S5 Fig. OR4D6 Diplotype.** This shows the intensity of Galaxolide against the diplotypes for OR4D6 including the two significant SNPs M263T and S151T. In the discovery cohort, the higher LD between these two SNPs does not allow enough resolution to see which SNP is driving the association. In the replication cohort, where there is lower LD, it appears that the T/T genotype of S151T is driving the Galaxolide anosmia phenotype.
(TIF)

**S6 Fig. Graphic abstract of the study.**
(TIF)

**S7 Fig. Effects of age and sex on odor intensity phenotype.** A) In the discovery cohort, several odors differ in intensity ranking between sexes and/or across age (Male = 1, Female = 2). B) In the validation cohort, there are no significant differences in odor intensity across sex. 3M2H intensity (at both concentration of the odor) significantly decreases as age increases after Bonferroni correction (p <0.002 for low concentration (dil1); p < 0.002 for high concentration (dil2)). For both cohorts, sex effects were tested using a t-test, and age effects were tested with a linear model. We used sex as a covariate in the linear model testing for age effects in the discovery cohort, but not the validation cohort, due to significant effects on sex in odor perception in the discovery cohort.
(TIF)

## Acknowledgments

We acknowledge the contributions from Young de Graaf from Unilever for making the odorant solutions in the discovery study; Marcia Knoop and the sensory panel from Unilever for isointensity testing; Tom Salmon from Unilever, for his help in selecting odorant stimuli and fragrances; and David Gunn from Unilever for helpful comments throughout the project.

## Author Contributions

**Conceptualization:** Fei-Ling Lim, Andreas Keller, Monique A. M. Smeets, Joel D. Mainland, Sijia Wang.

**Data curation:** Bingjie Li, Marissa L. Kamarck.

**Formal analysis:** Bingjie Li, Marissa L. Kamarck, Qianqian Peng.

**Funding acquisition:** Qianqian Peng, Andreas Keller, Joel D. Mainland, Sijia Wang.

**Investigation:** Bingjie Li, Marissa L. Kamarck, Qianqian Peng.

**Resources:** Monique A. M. Smeets, Joel D. Mainland, Sijia Wang.

**Visualization:** Bingjie Li, Marissa L. Kamarck, Qianqian Peng.

**Writing – original draft:** Marissa L. Kamarck.

**Writing – review & editing:** Bingjie Li, Marissa L. Kamarck, Qianqian Peng, Fei-Ling Lim, Andreas Keller, Monique A. M. Smeets, Joel D. Mainland, Sijia Wang.

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
