## [Decision Letter · Decision Letter 0]

22 Jun 2021

Dear Dr Wang,

Thank you very much for submitting your Research Article entitled 'From musk to body odor: decoding olfaction through genetic variation' to PLOS Genetics.

The manuscript was fully evaluated at the editorial level and by independent peer reviewers. The reviewers appreciated the attention to an important problem, but raised some substantial concerns about the current manuscript. Based on the reviews, we will not be able to accept this version of the manuscript, but we would be willing to review a much-revised version. We cannot, of course, promise publication at that time.

If you decide to revise the manuscript for further consideration at PLOS Genetics, please aim to resubmit within the next 60 days, unless it will take extra time to address the concerns of the reviewers, in which case we would appreciate an expected resubmission date by email to plosgenetics@plos.org.

[LINK]

We are sorry that we cannot be more positive about your manuscript at this stage. Please do not hesitate to contact us if you have any concerns or questions.

Yours sincerely,

Sarah Markt

Guest Editor

PLOS Genetics

Hua Tang

Section Editor: Natural Variation

PLOS Genetics

Reviewer's Responses to Questions

**Comments to the Authors:**

Reviewer #1: Review of Li et al.

The authors describe an extensive study of specific anosmias for a range of compounds initially in Han Chinese, finding their genetic basis through GWA and cell-based assays. Their specific discoveries are around the odorant receptor genes harbouring polymorphisms explaining differences in the intensity rating of musk compounds. This class of compounds is important as they are thought to potentially underpin certain behaviours in humans from kin to mate recognition. The authors also reconfirm a number of previously described associations between odorants and odorant receptor genes and also demonstrate that in the main loss of ability to smell many of these compounds is a derived state. These findings lead them to the conclusion that humans have been gradually increasing the number of segregating polymorphism for alleles that encode reduced ability to detect many odorants.

The authors have conducted a well thought out set of phenotyping regimes to compare with SNP data and have also replicated their finding in another population. And cell-based assays were successful in confirming one of the two new gene/odorant associations. The findings are significant to the field adding to our knowledge of the receptors underpinning musk perception. I have only a few suggestions on how the manuscript could be improved.

It would be good to add information on the origin and purity of the odorants used to the methods section.

Also it was not clear to me on the approach used in reusing data on the statistical treatment. Was an FDR approach or similar correction used for the GWA statistics?

The authors claim that this is the first undertaking of this type of study in Asian populations. McRae et al. (ref 34) did conduct a replication experiment within their 2013 study in SE Asians.

The authors might wish to consider also looking at the associations among the different polymorphisms across their study participants. How independent were they?

Minor points

The legend of Fig 1 has a sentence that needs attention. The fourth sentence … “ but not did examine” doesn’t make sense.

The last sentence of paragraph 1 on page 3 needs an “I” added to “In”

On page 11 in the Odor concentration and preparation section “difference” needs to be changed to “different”.

Reviewer #2: Li, Kamarck, Peng and colleagues examine the association between genetic variation in olfactory receptors and perception of 10 odors. They report improvements on previous methods (larger population size, a non-Western and homogenous study population, and inclusion of a validation study). Though the latter is a standard in the field, it is rarely successfully utilized for olfactory genetic studies. The authors discover novel associations for two interesting odors: a musk, which is a group of odors for which the coding of perceptual information has been difficult to determine, and a major contributor to body odor, which may have importance for human chemical communication. The results are of general interest for the study of large multi-gene families in which similar information, i.e. the “musk percept,” may be encoded by different receptors. Their analysis is statistically sound and the figures and tables are clear. Further clarification on a few questions is requested below:

1. Figure 1 does not indicate whether the association is with intensity or pleasantness, which might be interesting information to illustrate.

2. Given that the authors have phased haplotype information for OR4D6, it might be useful to add a figure (or a supplementary figure) on what galaxolide perception looks like for the different SNP combinations in their study populations. It appears the LD between M263T and S151T is different in the discovery and validation cohorts.

3. SI Tables S5 and S6 would benefit from including population frequency for these haplotypes.

4. The manuscript lacks a mechanistic explanation for the association with OR51B2, i.e. they fail to tie functional differences in this OR to differences in 3M2H perception. The associated variant, L134F, is found at a high frequency in the population in combination with other amino acid changes. Given the effect reported, one would hypothesize the F variant will increase receptor function. These variants should be tested or an explanation provided for why they weren’t examined.

5. There is a large difference between published data and what is reported here in the percent variation explained in beta ionone perception – 96% published for sensitivity and approximately 22% and 32% here for intensity. Could this have implications for coding of beta ionone information, i.e. more receptors are recruited for intensity perception?

6. In the introduction, the authors mention that one improvement to their methodology is using a non-Western and homogenous, i.e. Han Chinese population, for their discovery cohort. But this point is not mentioned later in the manuscript. The authors may want to consider adding a bit more on the benefit of studying a unique (non-Caucasian) cohort and whether its genetic similarity is beneficial to their discoveries. An interesting analysis (and associated figures) for quantifying genetic distance is discussed only in the methods section and illustrated in supplemental, so interested readers may miss these points. Along these lines, the distribution of 3M2H perceived intensity was very different among discovery and validation study populations. The authors note in the discussion that allele frequency for OR51B2 is similar among populations, meaning genetic differences cannot provide an explanation. Is this an example of diversity in study population being beneficial?

7. There is no discussion of the effect of age or sex on odor perception – were these examined with no effect or not considered in the study?

8. A brief sentence on some of the methods used would be useful. For example what constitutes the “credible set” or what a CMS score is (the acronym is undefined for its first use in the manuscript).

9. Why were different quality control cutoffs used for the discovery and validation cohort, i.e. a MAF of 1% for discovery and 5% for validation?

Reviewer #3: The authors conduct a GWAS using ten odors in 1003 Han Chinses and found novel associations for OR4D6 with the musk odor Galaxolide, and OR51B2 with 3M2H. Furthermore, they validate their results in a multi-ethnic independent population and different methodology, which demonstrates these associations are stable across populations and robust to differences in methods, including odor concentration and delivery method. Although the manuscript provides an interesting topic of public concern, there are several major concerns.

1. As shown in supplemental figure 1, some of the phenotypes are not normally distributed. The authors should provide more details on how they did normalization in association analysis. rank-based inverse normal transformation? log transformation?

2. The within-subject reliability of rating is acceptable in discovery cohort (test-retest r=0.75). However, the consistency in replication cohort is relatively low (test-retest r=0.69). I’m concerned that these subjective rating systems may bring bias results. Please mention this in the discussion/limitations.

3. The article has been submitted to a genetics journal but I suspect the current explanation of the methods may rely too much on familiarity with studies of Neuroscience. No doubt the authors can fix this. A graphic abstract may help the reader gain an overview of this study easily.

4. Is there a typo in ‘n both cohorts, all the genetic variants that were significantly associated with pleasantness were also significantly associated with intensity perception.’?

**Have all data underlying the figures and results presented in the manuscript been provided?**

Reviewer #1: Yes

Reviewer #2: Yes

Reviewer #3: Yes

PLOS authors have the option to publish the peer review history of their article (what does this mean?). If published, this will include your full peer review and any attached files.

Reviewer #1: No

Reviewer #2: No

Reviewer #3: No

---

## [Decision Letter · Decision Letter 1]

15 Oct 2021

Dear Dr Wang,

Thank you very much for submitting your Research Article entitled 'From musk to body odor: decoding olfaction through genetic variation' to PLOS Genetics.

The manuscript was fully evaluated at the editorial level and by independent peer reviewers. The reviewers appreciated the attention to an important topic but identified some concerns that we ask you address in a revised manuscript

We therefore ask you to modify the manuscript according to the review recommendations. Your revisions should address the specific points made by each reviewer.

[LINK]

Yours sincerely,

Sarah Markt

Guest Editor

PLOS Genetics

Hua Tang

Section Editor: Natural Variation

PLOS Genetics

Thank you for this detailed revision. There are a few remaining questions and minor revisions the reviewers have raised that we would like you to consider. These are outlined in the letter below.

Reviewer's Responses to Questions

**Comments to the Authors:**

Reviewer #1: I'm satisfied that the authors are made the necessary changes to the manuscript.

Reviewer #2: The authors responded thoroughly to all questions and suggestions for clarity! A few points:

1 – Thank you for providing the additional diplotype graphs! To clarify my earlier suggestion, I was interested in the effect of each haplotype on perceived intensity. Subjects are mainly reference (SM) or TT, although, as noted in your reply, this is more interesting in the validation cohort.

2 – Thank you for the additional work on the OR51B2/4 cluster. The legend for panel e indicates you tested OR51B2 L134F, but in the table describing tested variants you have the full haplotype (C120R;L134F;C209S). Did you test one amino acid change or all three? Also, please align with how the haplotype for OR51B4 is designated – changes in amino acid only vs. amino acid number.

3 – Line 45 needs a “1” in “OR5A1.”

4 – In the legend for Figure 3, is 0.02 the true value for the genomic lambda of galaxolide in the discovery cohort?

Reviewer #4: The manuscript by Li and colleagues reports a GWAS study where individual differences in the perception of a set of odors was found to be associated with genotypic variation. This reviewer was asked to evaluate the suitability of the revised manuscript for publication following a previous round of revisions that included three other reviewers. The following evaluation thus takes into account the author replies, as well as the revised manuscript.

All three previous reviewers posed insightful and in-depth comments on the present work, focusing mostly on methodological aspects related to genetic measurement and statistical assessment. Importantly, the authors provided thoughtful and elaborate replies to these comments, and the revisions were appropriate. Although I have identified some additional issues (listed below), they may be considered minor and relate mostly to the interpretation and presentation of the findings. I do not see any remaining major flaw in the revised version of the manuscript, and it is very well written overall.

Two of the main limitations of the present work are that (a) some odors for which there appears to have been strong a priori hypotheses (e.g. two aldehydes and MixA and MixB), did not give clear-cut results, which is hard to explain. This should be mentioned in the abstract. On the other hand, some novel results were discovered and validated across samples, they constitute significant contributions to the field and outweigh the limitations. Moreover (b) that odor perception assessments are somewhat different across samples. This makes discrepant results across samples difficult to interpret. These issues are not, in my view, obstacles to publication, but they should be highlighted and discussed to provide an appropriate context and transparency.

An issue posed in the prior evaluation is whether the test-retest reliability of odor ratings can be questioned. In my view, the reported reliability of these behavioral measures should be considered surprisingly high in the context of olfactory behavioral measurement. Numbers are also quite similar across samples. In fact, one would only expect higher behavioral test-retest correlations in samples deliberately selected to cover a very wide range of the performance spectrum (e.g. by over-sampling anosmics and hyposmics). High test-retest reliability of about 0.7 makes sense considering the odors are selected based on a highly variable sensitivity in the population – this would imply higher test-retest reliabilities compared to a situation where complex odors are presented at similar intensities. The rank-ordering of odors in each individual might potentially further increase the reliability, as might removing 4 participants who displayed no test-retest correlation. In sum, I find the behavioral assessment to be valid, but the authors may briefly note in the discussion how their methodological decisions led to these results.

The odor concentrations were established based on evaluations from 14 expert panelists, but these panelists are not well characterized (page 28). Who were they?

The authors state that ”the derived SNP alleles cause lower odor intensities”. Such statements should be revised to avoid strong causal inference from what is in principle correlational work, although mechanisms may be assumed. It could also be unpacked for a non-specialist readership - how is it determined that the SNPs are associated with lower, rather than higher, intensities. I assume this is based on the known evolutionary trajectories (the results contrasts odor intensity in variant > reference homozygotes), but this is not clearly explained (e.g. on page 10, in the introduction or abstract) until pages 15 and 37.

The authors report finding novel receptors for musk and a body odor compound. Please clarify briefly (also in abstract) how the association between odor perception and SNPs can only reasonably be explained by resulting variations in olfactory receptors because the targets were found on OR nuclei of the genome - as receptor activity was not directly studied, this would be helpful for a non-specialist to know in the early parts of the manuscript.

184-186 Explain the direction of the relationship of perceived intensity and pleasantness – how are these perceptual dimensions linked by genotypc variation (e.g. high intensity = low pleasantness)?

**Have all data underlying the figures and results presented in the manuscript been provided?**

Reviewer #1: Yes

Reviewer #2: Yes

Reviewer #4: Yes

PLOS authors have the option to publish the peer review history of their article (what does this mean?). If published, this will include your full peer review and any attached files.

Reviewer #1: No

Reviewer #2: No

Reviewer #4: No

---

## [Decision Letter · Decision Letter 2]

1 Dec 2021

Dear Dr Wang,

We are pleased to inform you that your manuscript entitled "From musk to body odor: decoding olfaction through genetic variation" has been editorially accepted for publication in PLOS Genetics. Congratulations!

Yours sincerely,

Sarah Markt

Guest Editor

PLOS Genetics

Hua Tang

Section Editor: Natural Variation

PLOS Genetics

Comments from the reviewers (if applicable):

Reviewer's Responses to Questions

**Comments to the Authors:**

Reviewer #1: I have carefully read all the reviewers comments and the authors responses as well as the revised manuscript. The manuscript has been significantly improved by the authors in response to these comments. As such I recommend the manuscript is ready for publication.

Reviewer #2: The authors replied to all comments and suggestions!

I agree with the authors that the diplotype graph is a better representation, but thank you for including the haplotype graph in your response.

Reviewer #4: The authors have responded to my previous comments in a thoughtful way and made appropriate revisions to the manuscript.

**Have all data underlying the figures and results presented in the manuscript been provided?**

Reviewer #1: Yes

Reviewer #2: Yes

Reviewer #4: **No: **The authors have limitations to access to genetic data at an individual level, but have shared GWAS summary statistics results. My understanding is that the data sharing is in compliance with the PLoS data policy, but this should be evaluated by the editors.

PLOS authors have the option to publish the peer review history of their article (what does this mean?). If published, this will include your full peer review and any attached files.

Reviewer #1: No

Reviewer #2: No

Reviewer #4: No

**Data Deposition**

http://datadryad.org/submit?journalID=pgenetics&manu=PGENETICS-D-21-00614R2

**Press Queries**

---

## [Editor Report · Acceptance letter]

15 Dec 2021

PGENETICS-D-21-00614R2 

From musk to body odor: decoding olfaction through genetic variation 

Dear Dr Wang, 

We are pleased to inform you that your manuscript entitled "From musk to body odor: decoding olfaction through genetic variation" has been formally accepted for publication in PLOS Genetics! Your manuscript is now with our production department and you will be notified of the publication date in due course.

With kind regards,

Livia Horvath

PLOS Genetics

On behalf of:
